# Exploring the Potential of Water-Soluble Squid Ink Melanin: Stability, Free Radical Scavenging, and Cd^2+^ Adsorption Abilities

**DOI:** 10.3390/foods12213963

**Published:** 2023-10-30

**Authors:** Shuji Liu, Xianwei Liu, Xueqin Zhang, Yongchang Su, Xiao’e Chen, Shuilin Cai, Dengyuan Liao, Nan Pan, Jie Su, Xiaoting Chen, Meitian Xiao, Zhiyu Liu

**Affiliations:** 1Fisheries Research Institute of Fujian, National Research and Development Center for Marine Fish Processing (Xiamen), Key Laboratory of Cultivation and High-Value Utilization of Marine Organisms in Fujian Province, Xiamen 361013, China; cute506636@163.com (S.L.); suyongchang@stu.hqu.edu.cn (Y.S.); caishuilin@hqu.edu.cn (S.C.); liaodengyuan@sina.com (D.L.); npan01@qub.ac.uk (N.P.); sjscut@126.com (J.S.); xtchen@jmu.edu.cn (X.C.); 2College of Chemical Engineering, Huaqiao University, Xiamen 361021, China; wynlxw@sina.com (X.L.); xqzhang2009@hqu.edu.cn (X.Z.); 3College of Food and Pharmacy, Zhejiang Ocean University, Joint Key Laboratory of Aquatic Products Processing Technology of Zhejiang Province, Zhoushan 316022, China; xiaoechen@163.com

**Keywords:** water-soluble squid ink melanin, stability, scavenging free radicals, adsorption of Cd^2+^

## Abstract

Squid ink melanin can be efficiently extracted from the byproduct ink sac generated during squid processing. As a natural food colorant, it possesses inherent antioxidant properties and the capability to adsorb heavy metals. This study aims to investigate the solubility of water-soluble squid ink melanin (WSSM) obtained from the ink sac, as well as its stability under various conditions including temperature, pH, salt, sugar, potassium sorbate, metal ions, sodium benzoate, sodium sulfite (reducing agent), and hydrogen peroxide (oxidizing agent). Moreover, it explores the scavenging effects of WSSM on free radicals and cadmium ions. The findings suggest that WSSM’s stability is insignificantly affected by high temperature, sucrose, and salt. However, acidity, sodium benzoate, potassium sorbate, sodium sulfite (Na_2_SO_3_), and hydrogen peroxide (H_2_O_2_) significantly influence its stability. Most metal ions do not impact the stability of WSSM, except for Fe^2+^, Fe^3+^, Al^3+^, and Cu^2+^, which result in the precipitation of WSSM. Additionally, WSSM exhibits remarkable antioxidant activity with IC_50_ values of 0.91, 0.56, and 0.52 mg/mL for scavenging superoxide anion radicals (O^2−^·), hydroxyl radicals (·OH), and DPPH radicals, respectively. It also demonstrates the ability to adsorb the heavy metal Cd^2+^, with the adsorption rate gradually increasing with a higher temperature and larger amounts of WSSM added. Infrared spectroscopy analysis reveals the weakening of characteristic peaks (-COOH and -OH) during the process of Cd^2+^ adsorption by WSSM, while SEM confirms surface roughening and structural damage after Cd^2+^ adsorption. This study provides valuable insights for the utilization of squid melanin products as natural antioxidants and heavy metal adsorbents in the food industry.

## 1. Introduction

Squid (*Todarodes pacificus*), also known as a soft fish or squid, belongs to the phylum Mollusca and class Cephalopoda [1]. Renowned for its delectable taste, it represents a high-protein, low-fat aquatic delicacy [2]. Abundant in essential elements, like calcium, phosphorus, and iron, it boasts remarkable edible and nutritional value [3]. Similar to other cephalopods, squid employs the ejection of dark ink as a defense mechanism [4]. Throughout squid processing, numerous byproducts emerge, including viscera, skin, cartilage, ink sac, and gonads [5]. Notably, the ink sac constitutes approximately 1.3% of the squid’s total weight. Predominantly composed of peptides, polysaccharides, trace elements, and melanin, the squid ink sac possesses an approximate melanin content of 20% [6]. This melanin variant belongs to the category of natural melanin and manifests as a polymer of indolequinone. Previous research has substantiated the potent biological activities of squid ink. These activities encompass anti-tumor [7], antibacterial [8,9], and anti-radiation effects [10,11]; the amelioration of lipid metabolism disturbances arising from cholesterol supplementation [12]; and immune regulation [10,13].

At present, with the increase in research on natural melanin, it is being widely applied in industries such as food, chemical, and light industries [14]. In the food industry, natural melanin is gradually replacing synthetic colorants due to its wide availability, safety, nontoxicity, and easy nutritional absorption [15]. It can be used as an additive in alcoholic beverages, beverages, and common food products [16]. The color of food is an important sensory quality [17]. Food colorants, as food additives, can be divided into two types: natural food colorants and artificial synthetic colorants [18]. Natural food colorants mainly come from microbial pigments, plant pigments, and mineral pigments extracted from animal and plant tissues. Natural food colorants have a high safety and nutritional value [19]. However, they are easily affected by factors such as metal ions, pH value, oxidants, light, and temperature, making them difficult to disperse, and they exhibit poor solubility between dyes and colorants [20,21]. Artificial synthetic colorants are mainly obtained through chemical synthesis. They have a bright color, stable performance, strong coloring power, high fastness, low cost, and are widely used. However, they may be toxic and cause side effects [22,23]. With the improvement in people’s living standards, healthy eating habits are receiving more and more attention. Therefore, the development and utilization of natural colorants are welcomed by people. At present, the market offers a wide variety of colorants, but natural melanin is relatively rare.

Squid ink melanin has attracted much attention because it is a kind of natural melanin and the only known endogenous biopolymer that can protect organisms from radiation damage [24], has attracted much attention. However, it can only dissolve in a few organic solvents and alkaline solutions, limiting its widespread application. Xin Guo et al. dissolved squid ink melanin under alkaline conditions, subjected it to ultrasound treatment, adjusted the pH of the solution to 7.0, performed desalination by dialysis, and obtained water-soluble squid ink melanin (WSSM) through vacuum freeze-drying methods [25]. Compared to squid ink melanin, water-soluble squid ink melanin can not only dissolve in alkaline solutions but also in water, making it more widely applicable. At present, there is limited research on water-soluble squid ink melanin. Lei Min et al. conducted a study on the bioactivity of water-soluble squid ink melanin and found that it can effectively improve lipid metabolism disorders caused by cholesterol supplementation. Water-soluble squid ink melanin can significantly protect the normal proliferation of H_2_O_2_-damaged endothelial cells and exhibits a certain degree of selectivity in inhibiting tumor cells [26]. However, there are currently no reported studies on the physicochemical properties of water-soluble squid ink melanin.

Water-soluble squid ink melanin (WSSM) is a complex biopolymer with defense and protection functions. Due to its special structure, WSSM can generate a large number of semiquinone free radicals, which can easily accept or donate an electron, making WSSM an excellent free radical scavenger [27]. In addition, WSSM also has strong metal cation adsorption properties, mainly through the action of anions such as carboxyl groups and deprotonated hydroxyl groups. At present, environmental pollution poses an increasing threat to human health, with frequent occurrences of environmental pollution incidents, especially cadmium contamination [28]. Once cadmium enters the human body, it forms cadmium-thiol proteins, which circulate throughout the body via the bloodstream and selectively accumulate in the kidneys and liver [29]. When accumulated to a certain extent, it causes irreversible damage to the human body [30,31]. Therefore, removing cadmium ions from wastewater is not only beneficial for environmental protection but also for human health. Cadmium can be separated and eliminated by methods such as solvent extraction, ion exchange, adsorption, precipitation, and electrolysis [32].

The ink sacs, as a byproduct of cephalopod processing, are generally discarded, resulting in a waste of resources, so extracting the melanin from cephalopod ink sac has a certain novelty, and can make full use of processing by-product resources. Therefore, the aim of this paper was to extract the melanin from the ink sacs of cephalopods, and explore the physicochemical properties of melanin and its ability to scavenge free radicals and adsorb cadmium ions, which would lay a certain experimental theoretical basis for the application of the melanin in medicine, the chemical industry, and the food industry. At the same time, it would also improve the added value of cephalopod processing, enhance the utilization capacity of marine animal resources, upgrade the technical content and competitiveness of products, and promote the development of China’s fisheries.

## 2. Materials and Methods

### 2.1. Materials and Reagents

Squid ink sacs were provided by Huifeng Food Industry Co., Ltd., (Putian, China). Analytical-grade reagents, including sodium bicarbonate, disodium hydrogen phosphate, sodium chloride, sucrose, sodium sulfite, lactic acid, sodium benzoate, petroleum ether, ferrous chloride, ferric chloride, zinc chloride, 37% hydrochloric acid, pyrogallol, and anhydrous ethanol were purchased from China National Pharmaceutical Group Chemical Reagent Co., Ltd., (Shanghai, China). Analytical-grade reagents, including sodium carbonate, citric acid, hydrogen peroxide, sodium lactate, ethyl acetate, cupric chloride, potassium chloride, aluminum chloride, calcium chloride, magnesium chloride, ferrous sulfate, salicylic acid, and cadmium chloride, were purchased from Xilong Chemical Co., Ltd., (Guangzhou, China).

### 2.2. Instruments and Equipment

The instruments and equipment used in this study included: a 752PC UV-visible Spectrophotometer, Shanghai Spectral Instrument Co., Ltd., (Shanghai, China); a 5804R centrifuge, Beijing Boyi Hengye Technology Development Co., Ltd., (Beijing, China); an SSW-600-2S electric constant thermostatic water tank, Medical equipment factory of Shanghai Boxun Industrial Co., Ltd., (Shanghai, China); a BP211D Electronic Balance, Sartorius Group, Germany; an FE-20-pH meter, Mettler-Toledo Instruments Co., Ltd., (Shanghai, China); an MS-2000 Laser Particle Size Analyzer, Malvern, UK; a SCIENTZ-10N Vacuum freeze Dryer, Ningbo Xinzhi Biotechnology Co., Ltd., (Ningbo, China); an AVATAR370 Fourier-transform infrared spectrometer, Thermo Company, (Waltham, MA, USA); and an S4800 scanning electron microscope, Hitachi Co., Ltd., (Tokyo, Japan).

### 2.3. Methods

#### 2.3.1. Preparation of the Water-Soluble Squid Melanin

Weigh 5.0 g of raw material and add it to a 1.18 mol/L NaOH solution in a liquid-to-solid ratio of 20:1 (*v*/*w*). Extract the mixture in a water bath at 40 °C for 1 h then centrifuge at 10,000 r/min for 20 min; then, collect the supernatant. Adjust the pH to 1.85 using HCl [33], and let it stand for 1 h. Centrifuge at 10,000 r/min for 15 min, take the precipitate and wash it 3 times. Subject the resulting insoluble squid melanin to vacuum freeze-drying. Dissolve the squid melanin by adding 1.18 mol/L NaOH solution, and treat it with an ultrasonic cell crushing apparatus for 1 h; adjust the pH to 7.0 with 6 mol/L hydrochloric acid, centrifuge for 15 min with 10,000 r/min, remove the precipitation, and retain the supernatant. After the dialysis and desalination of the supernatant, perform vacuum freeze-drying to obtain water-soluble squid melanin (WSSM) [25].

#### 2.3.2. Solubility of WSSM

Take 0.1 g of WSSM and place it in a 25 mL beaker. Add 10 mL of distilled water, anhydrous ethanol, acetone, ether, chloroform, sodium hydroxide solution, hydrochloric acid solution, petroleum ether, methanol, and ethyl acetate, respectively, at room temperature. Stir the mixture and let it stand for 10 min to observe the solubility and color of the solution.

#### 2.3.3. Particle Size Analysis of WSSM

The particle size distribution of the water-soluble squid melanin (WSSM) samples was measured using a laser diffraction particle size analyzer Beckman Coulter LS-230, (Beckman Coulter Co., Ltd., Brea, CA, USA). The WSSM was dispersed in deionized water. The particle size distribution (volume percentage) was generated using the Beckman-Coulter LS 230 instrument through the Beckman-Coulter LS 13320 software program [34].

#### 2.3.4. Thermal Stability Test of WSSM

Take 1 mL of a 5 mg/mL WSSM solution and place it in a 25 mL test tube. Dilute it up to 25 mL with distilled water and place it in constant-temperature water baths at 4, 25, 40, 60, and 80 °C, respectively, with distilled water as the reference solution. Measure the absorbance (A) of the solutions after 2, 4, 6, 8, and 10 h by ultraviolet spectrophotometer at 292 nm. The color change was observed and the retention rate of WSSM was calculated [35].
R (%) =100A/A0,(1)
where A is the absorbance value of the WSSM solution after a certain time, and A0 is the initial absorbance of the WSSM solution.

#### 2.3.5. Impact of Different Factors on WSSM Stability

##### pH Value

The buffer solutions with pH values of 3.0, 5.0, and 7.0 were prepared using a 0.1 mol/mL citric acid solution and a 0.2 mol/mL disodium hydrogen phosphate solution. Buffer solutions with pH values of 9.16 and 10.83 were prepared using a 0.1 mol/mL sodium bicarbonate and sodium carbonate solution, respectively. Subsequently, 1 mL of a WSSM solution with a concentration of 5 mg/mL was placed in a 100 mL volumetric bottle, diluted with a buffer solution of different pH values to 100 mL, and shaken well.

The absorbance value of WSSM at 292 nm was measured at 1, 2, 3, 4, 5, 6, and 7 days, with the corresponding buffer solution being used as a reference. The color change was observed. The WSSM retention rate was calculated using Equation (1).

##### Light Exposure

Take 2 parts of 1 mL of a 5 mg/mL WSSM solution and place them in separate 25 mL test tubes. Dilute each portion to 25 mL with distilled water and shake fully. Keep one portion protected from the light while exposing the other portion to the light at an intensity of 100 klx. After 1, 2, 3, 4, 5, 6, and 7 days, measure the absorbance of WSSM at 292 nm for both portions, observe any color changes, and calculate the retention rate of WSSM.

##### Salt

Prepare salt solutions with concentrations of 0.1, 1, 3, 5, and 10 mg/mL. Take 1 mL of a 5 mg/mL WSSM solution in separate 25 mL test tubes. Dilute each portion with the respective concentration of salt solution to a final volume of 25 mL, and thoroughly mix the solutions with the corresponding concentration of salt solution as the reference. After 1, 2, 3, 4, 5, 6, and 7 days, measure the absorbance of WSSM at 292 nm, observe any color changes, and calculate the retention rate of WSSM.

##### Sucrose

Sucrose solutions with concentrations of 0.01, 0.05, 0.1, 0.5, and 1 g/mL were prepared. A total of 1 mL 5 mg/mL WSSM solutions were placed in 25 mL test tubes, diluted to 25 mL with different concentrations of sucrose solutions, and thoroughly mixed, with the corresponding concentration of sucrose solution as the reference. The absorbance of WSSM at 292 nm was measured after 1, 2, 3, 4, 5, 6, and 7 days. Any color changes were observed, and the WSSM retention rate was calculated.

##### Preservatives

Sodium benzoate solutions with concentrations of 0.1, 0.5, 1, 2, and 5 mg/mL^,^ as well as potassium sorbate solutions with concentrations of 0.005, 0.01, 0.1, 0.5, and 1 mg/mL, were prepared. A total of 1 mL 5 mg/mL WSSM solution was placed in a 25 mL test tube. Various concentrations of sodium benzoate and potassium sorbate, serving as preservatives, were added to 25 mL separately, and shaken well, with the corresponding concentration of preservative solution as the reference. And the absorbance value of WSSM at 292 nm was determined after 1, 2, 3, 4, 5, 6, and 7 days, respectively. The color change was observed and the WSSM retention rate was calculated.

##### Oxidants and Reducing Agents

Hydrogen peroxide solutions with concentrations of 0.01, 0.1, 0.5, 1, and 2 mol/L were prepared as oxidizing agents. We placed 1 mL 5 mg/mL WSSM solution in a 25 mL test tube and added different concentrations of the hydrogen peroxide solution. The volume was adjusted to 25 mL, thorough mixing, with the corresponding concentration of the hydrogen peroxide solution without WSSM as the reference. The absorbance value of WSSM at 292 nm was measured at 1, 2, 3, 4, 5, 6, and 7 days; the color change was observed and the retention rate of WSSM was calculated.

Sodium bisulfite solutions with concentrations of 0.01, 0.1, 0.5, 1, and 1.5 mol/L were prepared as reducing agents. We placed 1 mL 5 mg/mL WSSM solution in a 25 mL test tube and added different concentrations of the sodium bisulfite solution. The volume was adjusted to 25 mL, thorough mixing, with the corresponding concentration of the sodium bisulfite solution without WSSM as the reference. The absorbance value of WSSM at 292 nm was measured at 1, 2, 3, 4, 5, 6, and 7 days; the color change was observed and the retention rate of WSSM was calculated.

##### Metal Ions

Solutions of Fe^2+^, Fe^3+^, Al^3+^, K^+^, Ca^2+^, Zn^2+^, Cu^2+^, and Mg^2+^ with concentrations of 50, 100, and 200 μg/mL were prepared. A total of 1 mL 5 mg/mL WSSM solution was placed in a 25 mL test tube, adding metal ion solutions with different concentrations. The volume was adjusted to 25 mL and thoroughly mixed, with the corresponding concentration of metal ion solution without WSSM as the reference. The absorbance value of WSSM at 292 nm was measured at 0 and 24 h; and the color change was observed and the retention rate of WSSM was calculated.

#### 2.3.6. Antioxidant Activity of WSSM

##### Determination of the Superoxide Anion Scavenging Capacity

We prepared phosphate buffer with a concentration of 0.1 mol/L (pH 8.34), absorbed it to 0.5 mL in a test tube, and added 1.0 mL distilled water; then, we placed the test tube in a constant-temperature water bath at 25 °C for 20 min. Afterward, a 0.5 mL preheated 10 mmol/L pyrogallol solution was added and mixed quickly; at the same time, we started the timer. Using 10 mmol/L hydrochloric acid as a blank control, the UV absorption value of the solution was measured at the wavelength of 325 nm, and the data were recorded 30 s later (*A*_1_), for every 30 s until 4 min (*A*_2_), and the autolysis rate SO_0_ of pyrocatechol was calculated [36].

Next, we absorbed 0.5 mL phosphate buffer with a concentration of 0.1 mol/L into a test tube, added 1.0 mL WSSM solution with different mass concentrations (0.1, 0.5, 1, 2, 3, 4, 5, and 6 mg/mL), and placed the test tube in a constant-temperature water bath at 25 °C for 20 min. Then, 0.5 mL preheated 10 mmol/L pyrogallol solution was added, mixed rapidly, and the timer started. A total of 10 mmol/L hydrochloric acid was used to replace pyrogallol solution as a control. Subsequently, the UV absorption value of the solution after sample addition was determined according to the above method, and the autooxidation rate of pyrogallol *SO*_1_ after sample addition was calculated.
(2)$$SO_{0}=\frac{A_{2}-A_{1}}{3}$$
(3)$$C=\frac{SO_{0}-SO_{1}}{SO_{0}}\times 100\%.$$

In Equations (2) and (3), *A*_1_ represents the UV absorbance value of the solution at 0.5 min. *A*_2_ represents the UV absorbance value of the solution at 4 min. *SO*_0_ represents the pyrogallol auto-oxidation rate before adding WSSM. *SO*_1_ represents the pyrogallol auto-oxidation rate after adding WSSM. C represents the clearance rate (%) of WSSM against O^2−^·.

##### Determination of the Hydroxyl Radical Scavenging Capacity

Concentrations of 9 mmol/L ferrous sulfate, 8.8 mmol/L hydrogen peroxide, and 9 mmol/L salicylic acid-anhydrous ethanol solution were prepared, and 0.5 mL was absorbed into the test tube successively. Different concentrations (0.2, 0.4, 0.6, 0.8, 1.0, 1.2, 1.4, 1.6, 1.8, and 2.0 mg/mL) of the WSSM solution (1.0 mL) were added to each test tube. After thorough mixing, the reactions were conducted in a constant-temperature water bath at 37 °C for 30 min. The absorbance of the solutions was measured at 510 nm, recorded as *A*. Using distilled water instead of H_2_O_2_, the absorption value of the solution was measured at 510 nm, denoted as *A*_0_; distilled water was used instead of WSSM solution as a blank control for determination at 510 nm wavelength, recorded as *A*_c_ [37,38].
(4)$$C_{1}=\frac{A_{c}-(A-A_{0})}{A_{c}}\times 100\%.$$

In Equation (4), *A* represents the absorbance value of the solution after adding the WSSM solution; *A*_0_ represents the absorbance value of the solution without hydrogen peroxide (H_2_O_2_) as the background absorbance; *A*_c_ represents the absorbance value of the blank tube; and *C*_1_ represents the clearance rate (%) of WSSM against ·OH (hydroxyl radicals).

##### Determination of the DPPH Free Radical Scavenging Capacity

Different concentrations (0.1, 0.5, 1.0, 1.5, 2.0, 2.5, 3.0, 3.5, 4.0, 4.5, and 5.0 mg/mL) of the WSSM solution (1.0 mL) were placed in test tubes. A 0.1 mmol/L DPPH solution was prepared in anhydrous ethanol, and 4.0 mL of this DPPH solution was placed in each test tube. After mixing and being left at room temperature for 25 min, the absorption value of the solution was measured at 517 nm, recorded as *A*. DPPH was replaced by anhydrous ethanol, and the absorption value of the measured solution was denoted as *A*_0_. Distilled water was substituted for WSSM solution as a blank, and the absorption value of the measured solution was denoted as *A*_c_ [39].
(5)$$C_{2}=\frac{Ac-(A-A_{0})}{A_{c}}\times 100\%.$$

In Equation (5), *A* represents the absorbance value of the solution after adding the WSSM; *A*_0_ represents the absorbance value of the solution without DPPH as the background absorbance; *A*_c_ represents the absorbance value of the blank tube; and *C*_2_ represents the clearance rate (%) of WSSM against DPPH free radicals.

#### 2.3.7. Adsorption of Cadmium Ions by WSSM [40]

##### Influence of Dosage on the Adsorption of Cadmium Ions by WSSM

A 10 mmol/L CdCl_2_ solution was prepared. A total of 50 mL CdCl_2_ solution was placed in a triangular flask with different amounts (5, 25, 50, 100, and 150 mg) of WSSM, shaken well, and placed at 30 °C for 1 h away from the light. Afterward, the solution was centrifuged at 10,000 rpm for 20 min and the supernatant was collected. The concentration of cadmium ions in the solution was determined using an atomic absorption spectrometer. The clearance (*CL*) efficiency of WSSM in adsorbing cadmium ions was calculated according to Equation (6).
(6)$$CL=\frac{C_{0}-C}{C_{0}}\times 100\%$$

In Equation (6), *CL* represents the adsorption rate (%) of cadmium ions; *C*_0_ represents the initial concentration of Cd^2+^; and *C* represents the concentration of Cd^2+^ in the solution after adsorption by WSSM.

##### The Influence of the Temperature on the Adsorption of Cadmium Ions by WSSM

A 10 mmol/L CdCl_2_ solution was prepared. A total of 50 mL CdCl_2_ solution was taken and placed into a triangular flask. Subsequently, 50 mg of WSSM was added to it, then shaken to avoid the light, and kept in water baths at different temperatures (25, 30, 35, 40, 45, and 50 °C) for 1 h. After centrifugation at 10000 r/min for 20 min, the supernatant was collected and the concentration of cadmium ions in the solution was measured using an atomic absorption meter. The clearance (*CL*) efficiency of WSSM in adsorbing cadmium ions was calculated according to Equation (6).

##### The Influence of Time on the Adsorption of Cadmium Ions by WSSM

A 10 mmol/L CdCl_2_ solution was prepared. A total of 50 mL CdCl_2_ solution was taken and placed into a triangular flask. Subsequently, 50 mg of WSSM was added to it, then shaken to avoid the light and kept in 30 °C water baths for different durations (30, 45, 60, 75, and 90 min). After centrifugation at 10000 r/min for 20 min, the supernatant was collected and the concentration of cadmium ions in the solution was measured using an atomic absorption meter. The clearance (*CL*) efficiency of WSSM in adsorbing cadmium ions was calculated according to Equation (6).

##### Infrared Spectroscopy Analysis of Adsorbed Metal Cadmium Ions by WSSM

After adsorbing cadmium ions, WSSM was subjected to vacuum freeze-drying. A trace amount of the dried material was placed in an agate mortar. An appropriate amount of potassium bromide was added, ground to a fine powder, and the pressed translucent sheet was then placed in a specially designed device in the infrared spectrometer for scanning to obtain the infrared spectrum information of the sample [41].

##### Scanning Electron Microscopy (SEM) Observation of Adsorbed Metal Cadmium Ion by WSSM

The powdered sample was dispersed in an ethanol solvent and subjected to microwave ultrasonic dispersion for 30 min. Then, it was dropped on the back of aluminum foil with a disposable straw, and the ethanol was blown with an ear ball. The location of the droplet was marked with a marker. After the droplet dried out, a suitable-sized piece was cut out and adhered to a sample holder with conductive adhesive for photography by a scanning electron microscope [42].

#### 2.3.8. Data Processing and Statistical Analysis

The data were processed and statistically analyzed using the Statistical Package for Social Science (SPSS 23.0 for Windows). The graphical representations were created using Origin 2021. All experiments were repeated in three operations.

## 3. Results and Discussion

### 3.1. Solubility and Particle Size Analysis of WSSM

WSSM was dissolved in different solvents, and it was observed that the WSSM solutions appeared black-brown. WSSM showed good solubility in water and alkaline solutions but had poor solubility in common acids and organic solvents, such as hydrochloric acid, methanol, chloroform, ethyl acetate, acetone, anhydrous ethanol, ether, and petroleum ether.

The size of particles is closely related to the specific surface area of an object [43]. Generally, smaller particle sizes result in larger specific surface areas, and the adsorption efficiency of adsorbents is influenced by their specific surface area [44]. In this study, a particle size and morphology analysis of WSSM was conducted using a particle size analyzer, and the results (Figure 1) indicated that WSSM had a uniform particle size distribution. The average particle size (D_50_) was determined to be 10.11 µm, with a specific surface area of 0.71 m^2^/g, which is significantly smaller than the average particle size of 17.34 µm reported by Xin Guo et al. [25]. According to Xin Guo et al.’s results, smaller particle sizes are associated with higher antioxidant properties. Thus, it can be predicted that the WSSM prepared in this study is likely to possess strong antioxidant activity.

### 3.2. Influence of Different Factors on the Stability of WSSM

Equal concentrations and volumes of WSSM solution were placed in water baths at different temperatures to investigate the impact of temperature and time on its stability. The results are shown in Figure 2A. As the temperature increased and the time was extended, there was little variation in the retention rate of WSSM, indicating that the temperature and time had a minor influence on its stability. Specifically, when WSSM was placed at 4 and 80 °C for 10 h, the retention rates were 98% and 97%, respectively. This suggests that WSSM had good thermal stability within the temperature range from 4 to 80 °C, and the color of the WSSM solution remained black and brown within this temperature range. The retention rate of WSSM under neutral and alkaline conditions with pH values of 7.0, 9.16, and 10.83, was high and almost unchanged, whereas under acidic conditions, with pH values of 3.0 and 5.0, the retention rate of WSSM was lower (Figure 2B). Additionally, it was observed that, when pH < 5.0, the WSSM solution began to form flocculent precipitates, causing the solution’s color to change from dark brown to clear and leading to a decrease in the absorbance values, resulting in a significant reduction in retention rate, which also indicates that acidic conditions have a significant impact on WSSM, making it less soluble in acidic solutions and more soluble in neutral and alkaline solutions, where its stability is less affected [45]. When the WSSM solution was exposed to the light at an intensity of 100 klx for 7 days, the retention rate remained at 92%, while the retention rate was still 96% after 7 days under the condition of no light (Figure 2C), suggesting that the influence of different light conditions on the stability of WSSM is relatively minor.

When water-soluble ink melanin is added as a pigment to food, it may come into contact with ingredients such as salt, sugar, preservatives, antioxidants, reducing agents, as well as metal ions from utensils or containers in food [46]. Therefore, the influence of these factors on the stability of WSSM was also monitored, and the results are shown in Figure 2D–I and Table 1. In Figure 2D, it can be seen that, as the concentration of table salt increased, there was relatively little change in the retention rate of WSSM. The retention rate of WSSM in the salt solution with 0.1 mg/mL for 7 days was 98.28%. When the concentration increased to 10 mg/mL, the retention rate remained at 97.02%, and the color of solution remained dark brown. This suggests that an increase in the salt concentration has a minor impact on the stability of WSSM. Similarly, as the concentration of sucrose increased, there was no significant change in the retention rate of WSSM. The retention rate was 98.57% after being placed in a sucrose solution with a concentration of 0.01 mg/mL for 7 days. When the concentration increased to 1 mg/mL, the retention rate was hardly affected, remaining at 97.28% (Figure 2E). The solution’s color also showed no significant change and remained dark brown, indicating that sucrose had a minor impact on the stability of WSSM.

Sodium benzoate and potassium sorbate are common food additives widely used as preservatives in food processing [47,48]. According to GB 2760-2014 (The National Food Safety Standard for Use of Food Additives), the usage levels of sodium benzoate and potassium sorbate are limited [49]. As shown in Figure 2F, with an increase in the concentration of sodium benzoate, the retention rate of WSSM showed a decreasing trend. Within the same concentration of sodium benzoate solution, the retention rate gradually decreased with longer storage times, and higher concentrations resulted in more significant decreases (*p* < 0.05). When the concentration of sodium benzoate was 0.1 mg/mL, the retention rate of WSSM remained above 90% after 7 days. However, when the concentration increased to 5 mg/mL, the retention rate dropped to 90.06% after 2 days and continued to decrease over time. By the 7th day, the retention rate of WSSM significantly decreased to 76.99% (*p* < 0.05). This indicates that high concentrations of the sodium benzoate solution have a significant impact on the stability of WSSM.

In Figure 2G, it can be seen that the retention rate of WSSM decreased with the increase in the potassium sorbate concentration. At low concentrations of 0.005 and 0.01 mg/mL, the retention rate after 7 days was not significantly affected, remaining above 90%. However, when the concentration exceeded 0.05 mg/mL, the retention rate of WSSM gradually decreased with a longer storage time. When the concentration increased to 1 mg/mL, the retention rate of WSSM significantly dropped to 71.99% after 7 days (*p* < 0.05), suggesting that a high concentration of potassium sorbate solution also has a significant impact on the stability of WSSM. Therefore, it can be concluded that preservatives have a minor impact on the stability of WSSM at lower concentrations, but at higher concentrations, they can significantly affect its stability. So, when it is used in combination, attention should be paid to its usage levels.

Airborne oxygen can react with substances in food, leading to the loss of nutrients [50]. Therefore, oxidants and reductants are commonly used in food processing. For example, oxidants are used for bleaching and whitening in products like bread, cakes, and biscuits, while reductants are used for preservation in products like meat and beverages [51,52,53]. Hence, the impact of oxidants and reductants on the stability of WSSM was also investigated. WSSM, due to its unique structure that can generate a large number of semiquinone radicals and easily accept or donate electrons, possessed certain antioxidant abilities. Therefore, it can be added as an antioxidant in food. In Figure 2H, it can be observed that the addition of different concentrations of hydrogen peroxide led to a decrease in the retention rate of WSSM, and the color of the solution gradually lightened. With an increase in the concentration of hydrogen peroxide, the retention rate significantly decreased (*p* < 0.05). When the concentration of hydrogen peroxide reached 2 mol/L, the retention rate of WSSM after 7 days was only 25%, and the solution became colorless, indicating that WSSM can act as an antioxidant because its reaction with hydrogen peroxide resulted in a reduction in the WSSM content and a fading of the solution’s color from black to colorless. When a reductant, sodium sulfite, was added, the retention rate of WSSM also gradually decreased, and the decrease became more pronounced with higher sodium sulfite concentrations. After 7 days at concentrations of 0.01 to 1 mol/L, the retention rate of WSSM remained above 90%. However, at a concentration of 1.5 mol/L, the retention rate significantly dropped to 86.54%, remaining at 86.50% after 7 days, although the color of the solution did not noticeably change and remained brown (Figure 2I). This suggests that sodium sulfite had a relatively minor impact on the stability of WSSM at lower concentrations but does affect its stability at higher concentrations.

According to Table 1, ions such as K^+^, Ca^2+^, and Mg^2+^ had no significant impact on the stability of WWSM, which was consistent with the findings of Zou Yu regarding the influence of metal ions on the stability of black auric melanin [37]. However, when WSSM came into contact with Fe^3+^, Fe^2+^, Al^3+^, and Cu^2+^, precipitation occurred, which may be due to the complex reaction of multiple o-phenol hydroxyl groups in the WSSM‘s molecular structure as multi-ligands with these metal ions. Two adjacent phenolic hydroxyl groups can form stable five-membered ring chelates with metal ions in the form of oxygen anions, leading to precipitation after complexation [54]. Therefore, when using WSSM, caution should be taken to avoid using it in conjunction with items containing Fe^3+^, Fe^2+^, Al^3+^, and Cu^2+^ ions.

### 3.3. Bioactivity of Water-Soluble Squid Melanin (WSSM)

#### 3.3.1. Antioxidant Activity of WSSM

Under alkaline conditions, pyrogallol causes an autoxidation chain reaction, releasing a significant amount of O^2^, which further accelerates the oxidation of phenol compounds, generating a series of intermediate products with characteristic absorption in the visible light region [55,56]. Antioxidant effectiveness can be assessed by adding antioxidants and monitoring changes in absorbance at 325 nm [57,58]. The inhibitory effect of WSSM on superoxide anion radicals (O^2−^·) is shown in Figure 3A. It could be observed from the graph that WSSM exhibited a strong capacity to scavenge O^2−^·. Within the range from 0.1 to 6 mg/mL, the clearance rate of O^2−^· by WSSM showed a linear upward trend. When the concentration was 6 mg/mL, the clearance rate of O^2−^· by WSSM reaches 100% and the IC_50_ value was 0.91 mg/mL. Compared to other commonly used antioxidants, WSSM had a lower ability to remove superoxide anion radicals.

Hydroxyl free radicals (·OH) are extremely reactive oxygen radicals with chemical properties and are also extremely harmful [37]. The Fenton reaction system can generate ·OH, which can selectively combine with salicylic acid to produce 2,3-dihydroxybenzoic acid [59,60]. UV spectrophotometry was used to detect the amount of the above-mentioned intermediate products to describe the ability of WSSM to remove ·OH. As shown in Figure 3B, WSSM exhibited a strong ability to clear ·OH. In the concentration range of 0.2~2 mg/mL, the clearance rate of ·OH by WSSM increased with the increase in the WSSM concentration. When the concentration was 2 mg/mL, the clearance rate of ·OH by WSSM reached 100% with an IC_50_ value of 0.56 mg/mL, which was comparable to the ability of squid melanin, as determined by Chen Shiguo et al. [61]. When compared to the common antioxidants studied by Liang Yun [62], it was found that WSSM was equivalent to Vc (IC_50_ = 0.56 mg/mL) and was higher than VE (IC_50_ = 1.21 mg/mL) and tert-butylhydroquinone (IC_50_ = 1.09 mg/mL) in terms of hydroxyl radical clearance ability.

Most free radicals are chemically reactive and have very short lifespans [63]. But a few free radicals have stable chemical properties and remain stable even at room temperature, with the DPPH radical (DPPH·) being one of them. Its stability mainly arises from resonance stabilization effects and spatial hindrance provided by three benzene rings, preventing the unpaired electrons on the central nitrogen atom from properly pairing [64]. The stability of DPPH· ensures the reliable measurement of a substance’s ability to scavenge free radicals [65]. The DPPH assay is simple, feasible, and stable, with good reproducibility and good universality, and less influenced by external factors [66]. DPPH radicals has a single electron, a strong absorption at 517 nm, with a purple color in alcohol solutions [67]. When a free radical scavenger is present, the absorption gradually disappears due to its single electron pairing, and the degree of fading is quantitatively related to the number of electrons accepted [68]. The inhibitory effect of WSSM on DPPH· is shown in Figure 3C. As can be seen in the figure, WSSM had a strong ability to scavenge DPPH·. The clearance rate of DPPH· by WSSM fluctuated and increased in the range of 0.1~5 mg/mL, with an IC_50_ value of 0.52 mg/mL. When the concentration was 5 mg/mL, the clearance rate of DPPH· reached 100%. Compared with other common antioxidants (Vc, IC_50_ = 13.5 μg/mL; VE, IC_50_ = 46.5 μg/mL, and lycopene, IC_50_ = 55 μg/mL), WSSM has a weaker DPPH· scavenging ability.

#### 3.3.2. Adsorption of Cadmium Ions by WSSM

As shown in Figure 4, with an increase in the WSSM supplemental level, the adsorption efficiency of Cd^2+^ significantly increased (*p* < 0.05). In the range from 5 to 25 mg, the adsorption rate of Cd^2+^ by WSSM showed a relatively minor increase. However, when the amount of added WSSM exceeded 25 mg, the adsorption rate of Cd^2+^ by WSSM rapidly increased. When the addition falls within the range of 100 to 150 mg, although the active sites of WSSM for the adsorption of Cd^2+^ continued to increase, the Cd^2+^ concentration in the solution is already relatively low at this time, resulting in a slow increase in the adsorption rate, reaching a plateau. The addition of 100 mg of WSSM led to a Cd^2+^ adsorption rate of 91.67%; further increasing the WSSM amount to 150 mg only increased the adsorption rate to 94.20%, representing a mere 4% increment. From both an adsorption efficiency and economic perspective, the optimal amount of WSSM was determined to be 100 mg.

In the study of the chemical adsorption of heavy metals, increasing the temperature enhances the adsorption efficiency of the adsorbent, because higher temperatures promote greater movement and collision chances between the adsorbent and metal ions. As shown in Figure 4B, the adsorption rate of Cd^2+^ by WSSM gradually increased with an increase of in the temperature. At 25 °C, WSSM exhibited an adsorption rate of 73.00%. When the temperature increased to 40 °C, the adsorption rate significantly increased to 83.80%, and further elevating the temperature to 50 °C resulted in a minor increase of 1.76%, reaching an adsorption rate of 85.55%, because the van der Waals bonds could be broken down when the temperature increase was high [69]. So, considering both adsorption efficiency and economic cost, a temperature of 40 °C was selected as the optimal operating temperature.

In the process of adsorbing metal ions, the longer the adsorption time, the more metal ions are adsorbed, but the process of the adsorption of metal ions is a dynamic equilibrium process where adsorption and desorption occur simultaneously [70,71]. When the system reaches equilibrium, the adsorption efficiency of the adsorbent on metal ions will remain within a small range of fluctuations [72]. As shown in Figure 4C, with increasing the time, there was little change in the adsorption rate of Cd^2+^ by WSSM. When the adsorption time was 30 min, the adsorption rate of Cd^2+^ by WSSM was 71.66%. Extending the adsorption time to 90 min resulted in an adsorption rate of 71.68%. It can be seen that an extended time had a minimal impact on the adsorption rate of metal ions. Gadd et al. studied the adsorption of Cd^2+^ in wastewater by bagasse and found that the adsorption equilibrium could be reached within 30 min [73]. Therefore, considering both adsorption efficiency and economic cost, a time of 30 min was chosen.

In addition, we analyzed the changes in the characteristic functional groups of WSSM before and after Cd^2+^ adsorption using infrared spectroscopy. The data showed (Figure 5A) that there was a wide and strong peak in the vicinity of 3500–3300 cm^−1^, mainly attributed to the stretching vibrations of -OH and -NH_2_ groups [74]. The peak at 2920 cm^−1^ corresponds to the stretching vibrations of alkyl C-H bonds [75], while the region between 1620 and 1550 cm^−1^ represents the stretching vibrations of C=C bonds in aromatic rings and the carboxylic group COO-C=O bonds [76]. The peak at 1220 cm^−1^ corresponds to the stretching vibrations of C-OH bonds in phenolic hydroxyl and carboxylic groups [77]. By comparing the infrared spectra of WSSM-adsorbed cadmium ions with those of WSSM before adsorption, it can be observed that the intensity of the peaks at 3400 cm^−1^ (-OH) [78] and 1514 cm^−1^ (-COOH) [79] decreased, indicating that both hydroxyl and carboxyl functional groups in the structure of WSSM may be involved in the adsorption reaction of WSSM with metal ions.

Furthermore, a comparative analysis of the apparent morphology of WSSM before and after Cd^2+^ adsorption was performed using scanning electron microscopy (SEM). The results (Figure 5B,C) show that, before Cd^2+^ adsorption, WSSM presented aggregated clusters with a smooth surface. After the adsorption of Cd^2+^, the surface morphology of WSSM was significantly altered, becoming rough and convex [80], suggesting that the structure of WSSM underwent a certain disruption upon Cd^2+^ adsorption. This finding is consistent with the results of Chen et al. [61], indicating that there was an interaction between WSSM and Cd^2+^.

## 4. Conclusions

As the demand for natural food colorants in the food industry continues to grow, squid ink melanin represents a natural black melanin product with stability, free radical protection, and heavy metal adsorption properties [21,81]. In this paper, water-soluble squid ink melanin (WSSM) was prepared from squid ink sacs through alkaline dissolution, acid precipitation, and subsequent desalination by vacuum freeze-drying. WSSM was found to be soluble in water and alkaline solutions but insoluble in acids and organic solvents. The particle size analysis revealed that the average volume particle size of WSSM was 10.11 µm using a particle size analyzer. High temperatures, sucrose, and table salt had minimal effects on the stability of WSSM. In contrast, acids, sodium benzoate, potassium sorbate, the reducing agent sodium bisulfite (Na_2_SO_3_), and the oxidizing agent hydrogen peroxide (H_2_O_2_) had significant impacts on the stability of WSSM. Most metal ions had no effect on the stability of WSSM, but Fe^2+^, Fe^3+^, Al^3+^, and Cu^2+^ caused the precipitation of WSSM. Meanwhile, the biological activities of WSSM were investigated, revealing its excellent scavenging effects on superoxide anions (O^2−^·), hydroxyl radicals (·OH), and DPPH·. The IC_50_ values were determined to be 0.91, 0.56, and 0.52 mg/mL, respectively, indicating that WSSM has a certain antioxidant activity. The influence of different factors on the adsorption of Cd^2+^ using WSSM was also examined. The results demonstrate that the adsorption of Cd^2+^ increased with higher temperatures and a greater dosage of WSSM, while the adsorption time had a minor impact. Moreover, the infrared spectroscopy analysis of WSSM before and after Cd^2+^ adsorption revealed a reduction in the characteristic peaks of -COOH and -OH, suggesting the involvement of these functional groups in the adsorption of WSSM with metal ions. Scanning electron microscopy (SEM) observations of WSSM’s morphological changes before and after Cd2+ adsorption confirmed that the surface of WSSM became rougher and underwent structural damage after adsorption.

In summary, water-soluble squid ink melanin (WSSM), as a natural pigment component, exhibited excellent stability and could serve as a new free radical scavenger and heavy metal adsorbent. Its high solubility can enhance absorption and utilization in the body, providing potential prospects for squid ink melanin products as natural antioxidants and heavy metal adsorbents. Furthermore, it has promising applications in the food processing industry.

## Figures and Tables

**Figure 1 foods-12-03963-f001:**
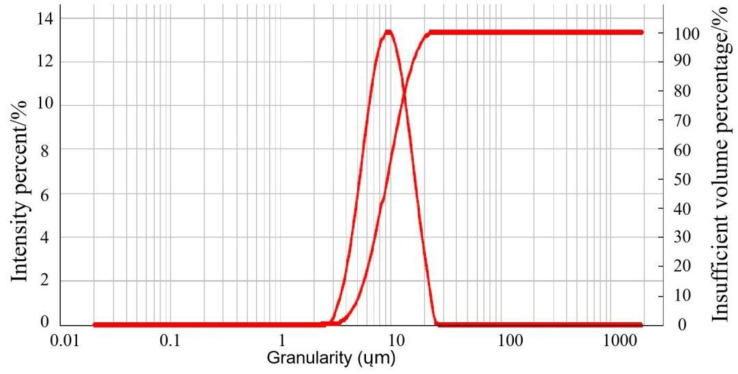
Particle size analysis of WSSM.

**Figure 2 foods-12-03963-f002:**
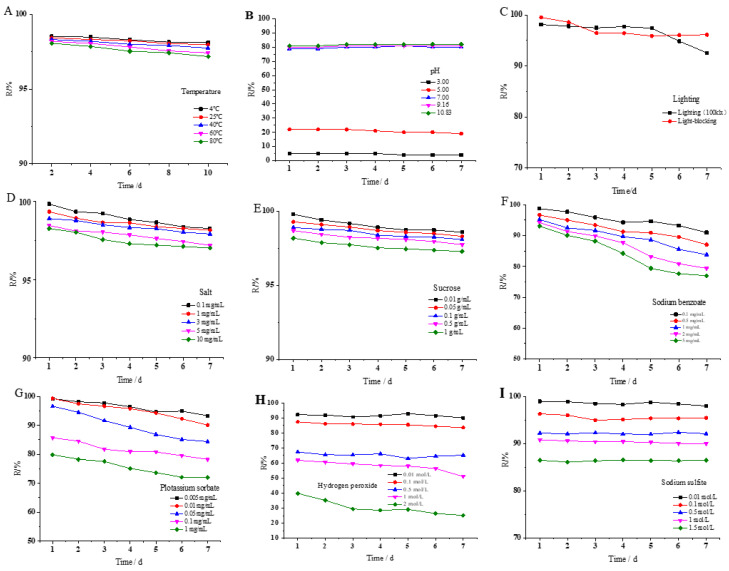
Effects of time and temperature (**A**), pH value (**B**), light (**C**), salt (**D**), sucrose (**E**), sodium benzoate (**F**), plotasstium sorbate (**G**), hydrogen peroxide (**H**), and sodium sulfite (**I**) on the stability of WSSM.

**Figure 3 foods-12-03963-f003:**
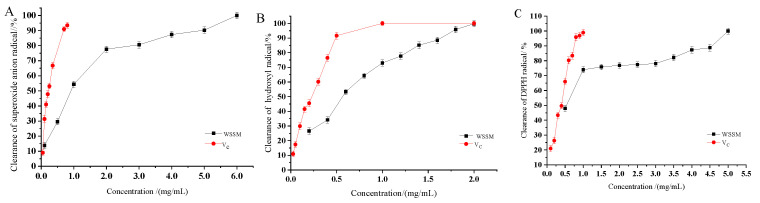
Scavenging effect of WSSM on superoxide anion radicals (**A**), hydroxyl free radicals (**B**), and DPPH radicals (**C**).

**Figure 4 foods-12-03963-f004:**
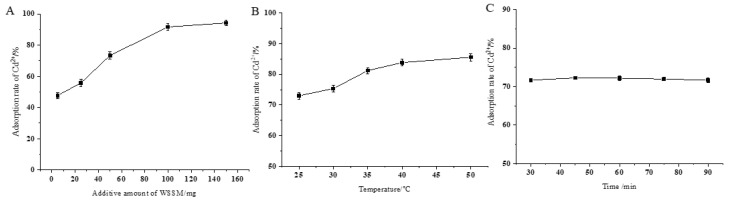
Effects of WSSM dosage (**A**), temperature (**B**), and time (**C**) on the absorption rate of Cd^2+^.

**Figure 5 foods-12-03963-f005:**
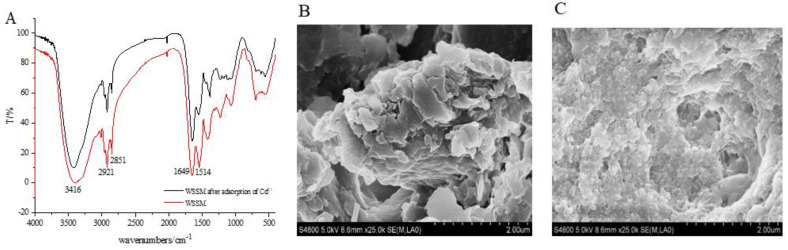
The IR spectra (**A**) and SEM images of WSSM before (**B**) and after (**C**) the adsorption of Cd^2+^.

**Table 1 foods-12-03963-t001:** Effects of different metal ions on the stability of WSSM.

Metal Ion	Ion Concentration μg/mL	The Retention Rate of WSSM/%	Color and State of the Solution
K^+^	50	100	—
	100	99	—
	200	84	—
Ca^2+^	50	99	—
	100	98	—
	200	96	—
Mg^2+^	50	95	—
	100	94	—
	200	92	—
Zn^2+^	50	87	—
	100	84	—
	200	83	—
Fe^2+^	50	—	Black precipitate
	100	—	Black precipitate
	200	—	Black precipitate
Fe^3+^	50	—	Yellow precipitate
	100	—	Yellow precipitate
	200	—	Yellow precipitate
Al^3+^	50	—	Yellow precipitate
	100	—	Yellow precipitate
	200	—	Yellow precipitate
Cu^2+^	50	—	Blue precipitate
	100	—	Blue precipitate
	200	—	Blue precipitate

## Data Availability

The data used to support the findings of this study can be made available by the corresponding author upon request.

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
