# Peer review of "Exploring the Potential of Water-Soluble Squid Ink Melanin: Stability, Free Radical Scavenging, and Cd2+ Adsorption Abilities"

_foods, 2023, doi:10.3390/foods12213963_

Round 1

Reviewer 1 Report

Comments and Suggestions for Authors

The manuscript "Exploring the Potential of Water-Soluble Squid Ink Melanin: Stability, Free Radical Scavenging, and Cd 2+ Adsorption Abilities" presents interesting research results. Melanins have a wide biological activity and may be used in food technology, both as antimicrobial and antioxidant agents, and as photoprotectants. Research into new applications of melanins is ongoing, as exemplified by the reviewed manuscript.

Detailed notes:

line 37 - italics in the name.

The entry of units is not consistent with the guidelines of the journal.

The methodology lacks a description of statistical methods for preparing the results. All results should be subjected to statistical analysis.

Figure 2, 4 - the authors should standardize the values on the OY axis.

Figure 3 - the authors should standardize the values on the OX and OY axes.

There is no information on how many repetitions the analyzes were performed.

Author Response

On behalf of my co-authors, we appreciate you very much for your positive and constructive comments and suggestions on our manuscript entitled “Exploring the Potential of Water-Soluble Squid Ink Melanin: Stability, Free Radical Scavenging, and Cd2+ Adsorption Abilities” (ID: foods-2661880). We have studied comments carefully and have made correction which we hope meet with approval. Revised portion are marked in red in the paper.

Reviewer 2 Report

Comments and Suggestions for Authors

All suggestions and comments are presented in the attached file.

Comments on the Quality of English Language

Minor editing of English language required

Author Response

(The authors gave the same response as above.)

Round 2

Reviewer 2 Report

Comments and Suggestions for Authors

Accept in present form